# Topologically frustrated ionisation in a water-ammonia ice mixture

C. Liu[1,2], A. Mafety[1], J.A. Queyroux[1], C. W. Wilson[3], H. Zhang[1], K. Béneut[1], G. Le Marchand[1], B. Baptiste[1], P. Dumas[4], G. Garbarino[5], F. Finocchi[6], J.S. Loveday[3], F. Pietrucci[1], A.M. Saitta[1], F. Datchi[1] & S. Ninet[1]

Water and ammonia are considered major components of the interiors of the giant icy planets and their satellites, which has motivated their exploration under high P–T conditions. Exotic forms of these pure ices have been revealed at extreme (~megabar) pressures, notably symmetric, ionic, and superionic phases. Here we report on an extensive experimental and computational study of the high-pressure properties of the ammonia monohydrate compound forming from an equimolar mixture of water and ammonia. Our experiments demonstrate that relatively mild pressure conditions (7.4 GPa at 300 K) are sufficient to transform ammonia monohydrate from a prototypical hydrogen-bonded crystal into a form where the standard molecular forms of water and ammonia coexist with their ionic counterparts, hydroxide ($OH^-$) and ammonium ($NH_4^+$) ions. Using ab initio atomistic simulations, we explain this surprising coexistence of neutral/charged species as resulting from a topological frustration between local homonuclear and long-ranged heteronuclear ionisation mechanisms.

[1] Institut de Minéralogie, de Physique des Matériaux et de Cosmochimie (IMPMC), Sorbonne Universités–UPMC Univ. Paris 6, CNRS UMR 7590, IRD UMR 206, MNHN, 4 Place Jussieu, F-75005 Paris, France. [2] Institute of Atomic and Molecular Physics and State Key Laboratory of Superhard Materials, Jilin University, Changchun 130012, China. [3] SUPA, School of Physics Astronomy Centre for Science at Extreme Conditions, The University of Edinburgh, Edinburgh EH9 3JZ, UK. [4] Synchrotron SOLEIL, Boîte Postale 48, 91192 Gif sur Yvette, France. [5] European Synchrotron Radiation Facility, Boîte Postale 2220, F-38043 Grenoble Cedex, France. [6] Institut des Nanosciences de Paris, Sorbonne Universités, UPMC Univ. Paris 6, CNRS UMR 7588, 4 Place Jussieu, F-75005 Paris, France. Correspondence and requests for materials should be addressed to F.D. (email: frederic.datchi@impmc.upmc.fr) or to S.N. (email: sandra.ninet@impmc.upmc.fr)

Knowing the properties of $H_2O$, $NH_3$ and their mixtures under high pressure and temperature is important for planetary science because these H-bonded ices are present in Jovian planets and their satellites[1–7] under a wide range of pressure (P) and temperature (T) conditions. The high P–T properties of the pure ice compounds have been the focus of many investigations, which have revealed a rich polymorphism. In particular, the existence of superionic water[8, 9] and ammonia[10], ionic ammonia[11, 12] and symmetric water[13–15] have been highlighted both experimentally and by theory[16–18]. The fast protonic diffusion in the superionic phase may be relevant to explain the unusual magnetic fields of Neptune or Uranus[19–25]. These phases only exist under extreme P–T conditions: $P > 60$ GPa and $T > 700$ K for superionic ammonia, $P > 47$ GPa and $T > 1000$ K for superionic water, $P > 100$ GPa and $P > 150$ GPa at 300 K for symmetric water ice and ionic ammonia, respectively.

By contrast with the pure components, information available on the dense phases of $H_2O$/$NH_3$ mixtures remain limited to modest pressures, although these mixtures are more relevant to the description of icy planets. They crystallise at low temperature into three different hydrates: ammonia monohydrate ($H_2O$:$NH_3$, noted AMH), ammonia hemihydrate ($H_2O$:$2NH_3$, noted AHH) and ammonia dihydrate ($2H_2O$:$NH_3$, noted ADH). Experimental studies on AMH have so far focused on the phase diagram below 10 GPa and low temperatures, where six different phases have been reported. Its phase diagram is depicted in Fig. 1a. The structures of phases III and IV are presently unknown, while those of I, II and V were solved in refs. [26–28], respectively. Phase V was actually found to be a mixture of the AHH-II and $H_2O$-VII phases resulting from the dehydration of AMH upon compression at 300 K[28]. Phase VI[29], also referred as the disordered molecular alloy (DMA) phase, is unique among H-bonded systems since it has a fully random substitutional disorder of ammonia and water over the sites of a body-centred cubic (bcc) structure (see Fig. 1b). This DMA phase was later found in the other stoichiometric compounds at distinct P–T conditions[30–32]. More recently, two theoretical studies[33, 34] have predicted that AMH transforms into an ionic solid composed of $NH_4^+$ and $OH^-$ ions arranged in a tetragonal (P4/nmm) structure at 10 GPa as presented in Fig. 1c. This self-ionisation is similar to the one observed in pure ammonia above 150 GPa[11, 12, 18] but would occur at much lower pressure in the ammonia monohydrate. However, no experimental evidence of ionic AMH has been reported so far.

Here we investigate samples of AMH up to 40 GPa at low temperatures from 5 to 300 K by experiments and first-principles computer simulations. Spectroscopic measurements reveal the presence of $NH_4^+$ and $OH^-$ species above 7.4 GPa but, at variance with the density functional theory (DFT) predictions of refs. [33, 34], AMH does not fully ionise up 40 GPa, as we observe the persistent signature of $NH_3$ and $H_2O$ molecules. Our diffraction data show evidence of the presence of a dominant bcc phase, similar to DMA, with a minor addition of a second phase which can be assigned to the ionic P4/nmm structure. Our computational study reveals that the DMA phase, at 10 GPa, spontaneously but partially ionise, to form a mixed ionic-molecular alloy. Complete ionisation is topologically hindered by the substitutional disorder which makes the proton transfer from a water to an ammonia molecule dependent on the near-neighbour environment. This ionico-molecular state of AMH, which we baptise disordered ionico-molecular alloy (DIMA), adds to the list of unconventional forms of ice.

## Results

**Ionisation at mild pressure**. About 30 different samples were studied in the present work. In all our experiments, AMH samples were prepared using an equimolar (at 1% precision) liquid mixture rapidly frozen to liquid nitrogen temperature and cold compressed to pressures between 10 and 25 GPa before warming up to room temperature (RT) (see the "Methods" section for experimental details). The low-temperature (LT) compression prevents the dehydration of the AMH solid, which occurs at the liquid–solid transition at RT[28]. Figure 2 shows the Raman and infrared (IR) absorbance spectra collected at 10 and 12 GPa, respectively. A striking feature of both spectra is the presence of a well-separated and sharp band at 3700 cm$^{-1}$. This band has not previously been observed in the ambient pressure, low-T IR spectra of any of the hydrates[35, 36]. To determine whether this results from the predicted pressure-induced ionisation of AMH, we compared the experimental spectra to the theoretical ones computed for the P4/nmm ionic structure (see "Methods" section for computational details). As seen in Fig. 2, the calculations do predict a Raman and IR active band at 3745 cm$^{-1}$ in the P4/nmm structure, i.e., within 1.2% of the observed one, which originates from the stretching of the $OH^-$ ions. In addition, the experimental spectra present Raman and IR activity near all the predicted frequencies for the P4/nmm structure, which include the lattice bands in between 200 and 800 cm$^{-1}$, the $NH_4^+$ bending (1460 and 1540 cm$^{-1}$), twisting (1840 cm$^{-1}$) and stretching (2715 and 2970 cm$^{-1}$) vibrations.

This correspondence would suggest that AMH at 10 GPa has indeed transformed into the P4/nmm ionic structure. However, the experimental spectra also show spectral bands in frequency ranges where none is predicted for P4/nmm: the Raman bands peaked at 1630, 2040 and 3300 cm$^{-1}$, and the IR bands at 980 and

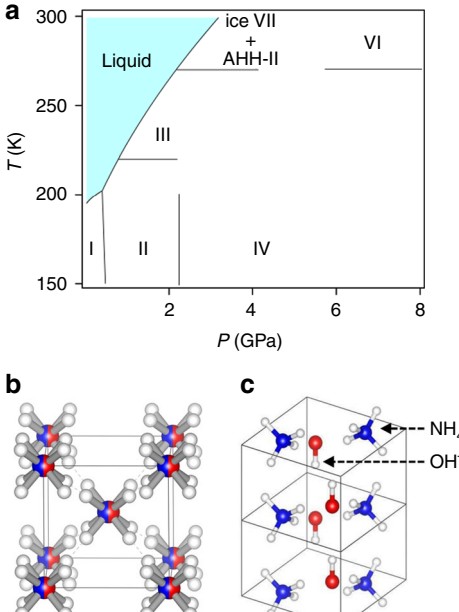

**Fig. 1** Phase diagram and structures of AMH. **a** Phase diagram of AMH. **b** Representation of the $Im\overline{3}m$ structure of the DMA phase VI[26]. The oxygen and nitrogen atoms are represented with the blue and red spheres, respectively. The water and ammonia molecules occupy the same sites with a 50% probability. The white spheres show the 16 possible positions along the ⟨111⟩ direction for the five hydrogen atoms in the unit cell. The other possible sites for H atoms along ⟨110⟩ are less probable and not represented for clarity. **c** Representation of the predicted ionic P4/nmm structure according to ref. [33]. The same colour code is adopted for the atoms

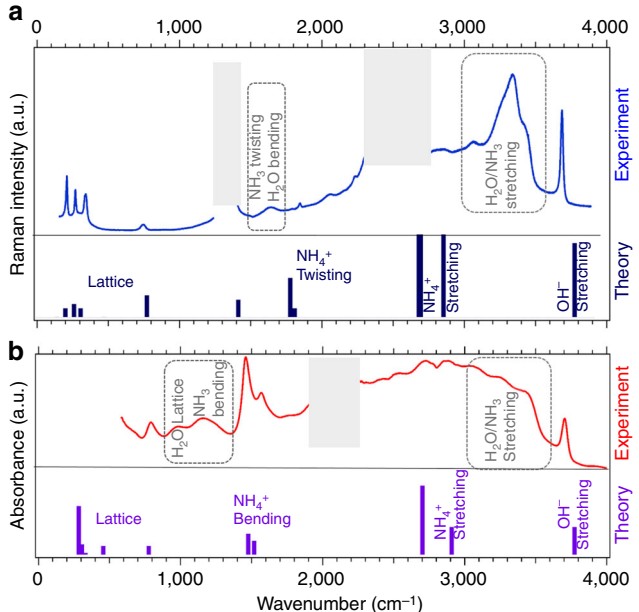

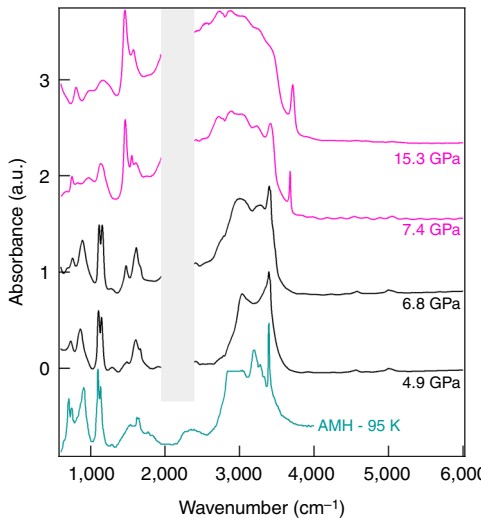

**Fig. 3** Evolution with pressure of the experimental IR absorption spectra of AMH. The spectra were collected upon decompression at RT. Pink and black curves are spectra collected above and below the ionic-molecular phase transition, respectively. The frequency window from 2000 to 2300 cm$^{-1}$ is obscured by the strong absorption band of the diamond anvils. Pressures are indicated on the right. The blue curve is the experimental spectrum of the molecular AMH-I phase at ambient pressure and 95 K from ref. [35]

**Fig. 2** Vibrational spectra of AMH above 7.4 GPa. **a** Raman and **b** infrared absorption spectra of AMH, at RT and respective P of 10 and 12 GPa. The experimental spectra (collected at RT) and the theoretical ones (calculated at 0 K) for the fully ionic $P4/nmm$ structure are shown in the upper and lower panels, respectively. In the experimental Raman spectra, the frequency windows from 1300–1400 cm$^{-1}$ to 2200–2600 cm$^{-1}$ are greyed as they are dominated by, respectively, the first- and second-order Raman signal from the diamond anvils. In the experimental infrared absorption spectrum, the frequency window from 2000 to 2300 cm$^{-1}$ is obscured by the strong absorption band of the diamond anvils. For visibility, Raman (respectively, IR) simulated intensities have been divided by 30 (respectively, 5) for NH$_4^+$ and OH$^-$ stretching modes

1150 cm$^{-1}$. The broad IR band extending from ~2300 to ~3500 cm$^{-1}$ can also hardly be assigned to the mere stretching modes of NH$_4^+$ predicted at 2715 and 2900 cm$^{-1}$. The frequencies of these "extra" bands are in turn characteristic of the vibrations of the H$_2$O and NH$_3$ molecules, which are observed in the ambient pressure hydrates[35, 37]. The Raman stretching band at 3300 cm$^{-1}$ is broad with little features, which is more typical of the disordered solid phases in the pure ices[38, 39]. The Raman and IR spectra thus suggest that the sample is a mixture of ionic and molecular species. We also observed that the intensities of the Raman peaks assigned to the $P4/nmm$ ionic structure depend on the position of the laser spot (of size 2–3 µm) on the sample and are anti-correlated to those assigned to the molecular species, suggesting that the distribution of ionic species is not homogeneous over the size of the sample. Moreover, these peaks varied in intensity between different loadings (Supplementary Fig. 3 shows the Raman spectrum of a different sample), and could not be observed at all in some samples.

The evolution of the IR spectra with pressure was followed during decompression at RT, examples of collected IR spectra are shown in Fig. 3 and Supplementary Fig. 1. From 40 down to 7.4 GPa, the spectra are all similar to that of Fig. 2. The frequency of the modes assigned to the $P4/nmm$ structure correlate very well with the predicted ones over the full pressure range (see Supplementary Fig. 2). The ratio between molecular and ionic species appears not to be much affected by pressure as the relative intensity of the respective IR bands do not change with pressure. Below 7.4 GPa, large modifications of the IR absorbance occur. Notably, all the spectral features correlated with the ionic species, in particular the OH$^-$ stretching band at ~3700 cm$^{-1}$ and the

NH$_4^+$ bending at 1460 cm$^{-1}$, disappear below this pressure, and only those assigned to the neutral H$_2$O and NH$_3$ molecules remain. Moreover, the widths of most spectral bands decrease below 7.4 GPa, and previously unresolved features appear. This is strongly indicative of a phase transition to a purely molecular phase. We also note that the IR spectrum at 7.4 GPa contains both features from the high-pressure and low-pressure spectra, suggesting a coexistence of the two phases and thus a first-order transition. IR spectra were also collected from 10 to 30 GPa at 100 K, showing no appreciable change of the IR absorbance with temperature between 300 and 100 K. The high-pressure phase is thus stable over a large range of P–T conditions, which is confirmed by our X-ray diffraction experiments as discussed hereafter. Our spectroscopic observations thus point to a partial ionisation of AMH above 7.4 GPa. Although several spectral bands match those expected for the $P4/nmm$ structure, the latter is a fully ionic structure and does not explain the presence of the spectral bands originating from H$_2$O and NH$_3$ molecules. To understand the structure of the high-pressure phase, we conducted X-ray and neutron diffraction experiments as described below.

**Substitutional and orientational disorder**. Angular dispersive X-ray diffraction (XRD) experiments were conducted either at beamline ID27 of the European Radiation Synchrotron Facility or using an in-house diffractometer (for details, see "Methods" section). We first present the results obtained at 10.6 GPa and RT on the sample whose Raman spectra is shown in Fig. 2. This sample of diameter 100 µm was mapped by XRD in steps of 10 µm, taking advantage of the small X-ray beam (3 × 3 µm FWHM) of the ESRF-ID27 beamline. The XRD images show that the sample is a textured powder. Five to six Bragg reflections are observed at all sample positions, which can be indexed by a bcc unit cell with lattice parameter $a = 3.322$ Å. This is the same unit cell as that found for the AMH-VI (DMA) phase in ref. [29]. A second set of reflections is also present with variable intensities depending on the sample position. These reflections can be

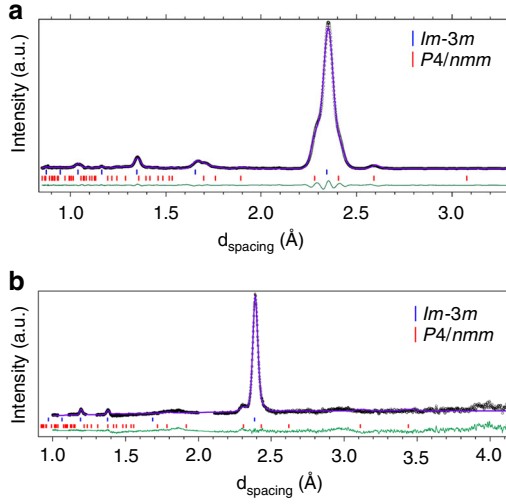

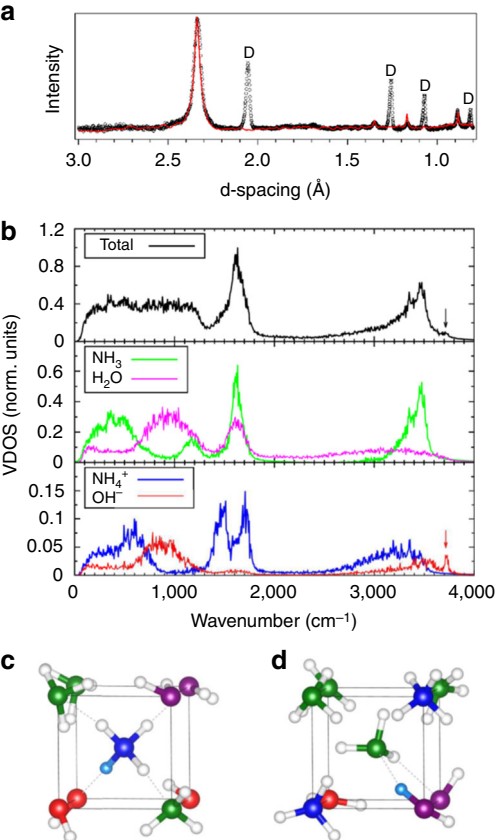

**Fig. 4** Diffraction patterns of AMH above 7.4 GPa. **a** X-ray and **b** neutron diffraction patterns of AMH, at RT and respective P of 10.3 and 8.6 GPa. The symbols are experimental data after subtraction of background and removal of the reflections from the diamond anvils in the neutron pattern. The purple solid lines are Le Bail (**a**) or full profile Rietveld (**b**) refinements using a mixture of $Im\overline{3}m$[29] and $P4/nmm$[33] structures. The green line shows the difference between observed and calculated profiles. Sticks show the positions of Bragg reflections

indexed by the $P4/nmm$ ionic structure with $a = 4.821$ Å and $c = 3.08$ Å, which is close to the predicted cell parameters for this structure at the same pressure, $a = 4.861$ Å and $c = 3.024$ Å (see Supplementary Table 1 for the complete list of structural parameters). Figure 4a shows the integrated XRD pattern of one image and the Le Bail fit using a mixture of $Im\overline{3}m$ and $P4/nmm$. In several other samples, the peaks from $P4/nmm$ were more difficult to detect, an example of which is shown in the Supplementary Fig. 4. This suggests that the high-pressure solid is mainly composed of the DMA phase, with a variable but minor addition of $P4/nmm$.

Additional information on the structure of high-pressure AMH was sought using time-of-flight powder neutron diffraction (ND) at the PEARL station of the ISIS facility (see "Methods" section for details). A deuterated sample was used in this case, since hydrogen has a strong incoherent neutron cross-section. The sample was loaded and compressed to 6.5 GPa at 150 K in order to obtain the AMH-VI phase as described in ref. [29], and then compressed from 6 to 12 GPa at room temperature. No obvious change of the diffraction pattern was detected and the structural model of the DMA phase gives a good fit of the data in the whole pressure range. No evidence of the $P4/nmm$ structure was found in this experiment, as seen in Fig. 5a. In a second loading though, non-bcc peaks were observed around the (110) peak of $Im\overline{3}m$ at 8.6 GPa, which can be interpreted as the (200) and (111) peaks of $P4/nmm$, as shown in Fig. 4b.

The P–T phase diagram of AMH was further mapped using XRD. Consistently with our IR spectroscopy measurements, we did not observe any phase transition on cooling the sample from 300 to 50 K at 12–13 GPa, or on compressing the sample up to 40 GPa at RT. On decompression at RT, the Bragg reflections of the bcc phase suddenly broadened and took the appearance of unresolved multiplets at 7.2 GPa. The sample became homogeneous again at 5.1 GPa, both visually and by XRD. The new Bragg reflections appearing below 7.2 GPa are much sharper than those of AMH-VI. Most of them can be indexed by the AHH-II structure reported by ref. [28] but not all. At 4 GPa, the

**Fig. 5** Structural and vibrational properties of the disordered ionico-molecular alloy. In **a**, we compare the simulated neutron pattern for the 6 × 6 × 6 simulation box (red line) to the experimental neutron pattern collected at 10 GPa, 295 K (symbols). The x scale of the simulated pattern has been multiplied by 0.994 to account for the density difference, and the intensity adjusted to scale with the main (110) peak. The Bragg peaks were modelled by pseudo-Voigt profiles of constant width. "D" indicates the reflections from the diamond anvils. **b** shows the vibrational density of states obtained from the AIMD trajectories. The arrows indicate the OH⁻ stretching mode. **c**, **d** Represent simulation snapshots showing the nearest-neighbour environment of an ammonium ion (**c**) or an ammonia molecule (**d**). The H atom involved in the H-bond between the central ammonia and neighbour water molecule is rendered in cyan, the others are represented by white spheres. N atoms are in green and blue for $NH_3$ and $NH_4^+$, respectively, and O atoms are in magenta and red for $H_2O$ and $OH^-$, respectively. The solid and dashed lines depict the cubic unit cell and the H bonds, respectively. For clarity, all species have been placed on their average sites

XRD pattern changes again and is the same as that observed on freezing the 1:1 liquid at RT and interpreted as a mixture of AHH-II and ice-VII[28].

To sum up, both XRD and ND show that samples of AMH above 7.4 GPa are mainly composed of the bcc phase previously observed by ref. [29], but also contain the $P4/nmm$ phase as a minor addition. The observed variability in the amount of $P4/mnn$ phase between different samples is consistent with the variations in Raman and IR peak intensities noted above. It is likely due to differences between loadings in the compression and warming rates, the pressure at which the sample is warmed up to RT, and/or to small deviations (below 1%) from the 1:1 composition (see "Methods" section). Indeed, as seen hereafter, the creation of ionic species is strongly dependent on the local environment, and the final state is controlled by kinetics.

**The disordered ionico-molecular crystal**. Our experimental results confirm the predicted existence of a $P4/nmm$ ionic structure of AMH above 7.4 GPa at RT, as its signature is observed in both spectroscopic and diffraction data. However, the dominant phase in our samples is always the $Im\overline{3}m$ structure found earlier by Loveday et al. This coexistence of a fully ordered ionic crystal, on one hand, and of a disordered molecular alloy, on the other hand, is highly intriguing and raises the following questions: first, if $P4/nmm$ really is the thermodynamically stable state, why is it only present at best as a minor phase in all our samples? Second, is the $Im\overline{3}m$ phase purely molecular?

We recall that $P4/nmm$ emerged as the lowest enthalpy structure at 10 GPa–0 K in theoretical calculations using the ab initio random structure search (AIRSS) code[40], where searches spanned unit cells with up to 8 $NH_3 \cdot H_2O$ formula units[33]. We used the same code to perform new searches in a larger pressure range, from 10 to 40 GPa, with up to 4 f.u. per unit cell (see "Methods" section for details on the search strategy). The result is that $P4/nmm$ is indeed the lowest enthalpy structure coming out from all searches in this pressure range, which also agrees with the recent report of ref. [34] who used a different search strategy and code. It is clear, however, that these searches are not able to find the $Im\overline{3}m$ structure, which fits present and previous[29] diffraction data for the bcc phase, because the large disorder of this structure cannot be described with the small numbers of f.u. to which the searches are limited. As a matter of fact, this disorder precludes a direct description of the $Im\overline{3}m$ structure in theoretical calculations, and thus to discuss its stability with respect to $P4/nmm$.

Griffiths et al.[33] noted that an ionic equivalent of Loveday et al.'s $Im\overline{3}m$ phase could be constructed from the $P4/nmm$ structure by (1) slightly contracting the $a$ and $b$ axes to make the unit cell cubic, (2) sliding the O atom along $z$ by $0.1562c$ to bring it to the centre of the cube and (3) mixing the occupancies of the two bcc sites with ammonium and hydroxyl ions to obtain the substitutional disorder imposed by the $Im\overline{3}m$ structure. If we assume that these operations would present a gain in energy with respect to $P4/nmm$, this could explain why the $Im\overline{3}m$ phase is dominant, but the latter would then not be molecular as previously assumed by Loveday et al.[29], but fully ionic. We confronted this structural model (The resulting structure, of space group $Im\overline{3}m$, has N and O atoms positioned at (0, 0, 0) with 50% probability, and H (or D) atoms on $(x, x, x)$ and (0, 0, z) (where $x$ and $z$ are chosen to have N-H(D) and O-H (D) distances of 1.04 and 0.97 Å, respectively) with occupancy factors of 1/4 and 1/12, respectively.) to the neutron diffraction data at 6 GPa and found that it fits almost equally well the diffraction pattern than the original molecular $Im\overline{3}m$ structure (the respective weighted R-factors are 2.63 and 2.61%). This disordered ionic structure however presents a major flaw: the substitutional disorder implies that two ammonium ions have a 25% probability to be first neighbours, i.e., distant by ~2.85 Å. The corresponding Coulomb repulsion would have a large energetic cost, which makes it very unlikely that such a structure would be stable. This reasoning leads us to think that the bcc phase cannot be fully composed of ionic species and thus remains partly molecular, which also complies with the observed signature of both ionic and molecular species in the vibrational spectra.

To go further and achieve a better understanding of the $Im\overline{3}m$ phase, we performed an extensive theoretical study of the latter and discuss the results below (for the complete description of the theoretical methodology, see the "Methods" section). First, we constructed an approximate model of this structure, compatible with computational capabilities, by generating a few tens of supercells containing $4 \times 4 \times 4$ bcc units, where 64 water and 64 ammonia molecules were distributed at random over the crystal

sites according to the $Im\overline{3}m$ structure of ref. [29]. We also constructed one $6 \times 6 \times 6$ supercell containing 432 molecules to check for size effects. Those cells were annealed using classical force fields, in order for the molecules to achieve the most stable orientational configuration. The 10 most stable $4 \times 4 \times 4$ cells and the $6 \times 6 \times 6$ cell were then selected, and optimised by DFT relaxation at 0 K. All those structures displayed some ionisation, characterised by a proton relaxation from one donor water molecule to an acceptor ammonia one to which it is H-bonded, and thus resulting in a substitutionally disordered crystal where $NH_4^+$ and $OH^-$ ions coexist with $NH_3$ and $H_2O$ neutral molecules. This ionisation concerned 12–17% of the molecules in the $4 \times 4 \times 4$ cells, and 13% in the $6 \times 6 \times 6$ one, showing that the concentration of ionic pairs does not depend much on the cell size. Figure 5a compares the neutron pattern computed from the atomic positions of the $6 \times 6 \times 6$ supercell to the experimental one at 10 GPa (see Supplementary Fig. 5 for the X-ray pattern). The excellent agreement shows that, despite the additional disorder induced by ionisation, the average structure remains of $Im\overline{3}m$ symmetry and that our model is compatible with experiment.

For comparison, we also built one supercell with an ordered version of the structure, i.e., where one of the two "bcc" sites was systematically occupied by a water molecule, and the other one by an ammonia one. In this case, the $T = 0$ K relaxation immediately resulted in a full ionisation of the system in the $P4/nmm$ structure. These results strongly suggest that the ionisation process in the substitutionally disordered structure depends on the local arrangement of species and topology, which will be further analysed below.

In order to understand the effects of temperature, ab initio molecular dynamics simulations (AIMD) was performed at 300 K at a pressure of 10 GPa on one supercell from the above set, which contained 12% ionic species after 0 K optimisation. These simulations show that the sample undergoes a further, dynamically induced ionisation: after ~8 ps, the number of ionic species reached the maximum values of 36% f.u. of $NH_4 \cdot OH$, a ratio which remained stable up to ~20 ps. The vibrational density of states (VDOS) obtained from these simulations are shown in Fig. 5b. As expected, vibrational modes from both molecular and ionic species are present, and the decomposition of the full VDOS (top panel) into their respective contributions are shown in the middle and lower panel. Of particular interest is the higher energy mode at 3715 cm$^{-1}$: the projected VDOS clearly shows that it comes from the $OH^-$ stretching, as in $P4/nmm$, however, its intensity in the total VDOS is much weaker in the disordered bcc phase than in $P4/nmm$ due to the relatively low concentration of about 7.5% $OH^-$ vibrators. The same is true for the $NH_4^+$ vibrational bands around 1500 cm$^{-1}$. As a matter of fact, the vibrational bands of the ionic species in the bcc phase are barely discernible from those of molecular species, which confirms that the Raman and IR bands assigned to ionic species in the experimental spectra of Fig. 2 mainly come from the occurrence of $P4/nmm$ in our samples.

We next try and explain why the ionisation is only partial in the bcc structure. To illustrate the main result of our analysis, we show on one hand in Fig. 5c a local configuration where a $OH^-$–$NH_4^+$ pair has formed through a proton transfer from the water molecule at the corner to the ammonia molecule at the centre of the cube to which it is H-bonded. On the other hand, Fig. 5d shows a configuration where no proton transfer has occurred between an equivalent pair of molecules. The main difference between these two configurations is that in the second case, other ammonium ions which formed at an earlier instant are first neighbours of the central ammonia molecule. In fact, the full substitutional disorder of ammonia and water units over the crystal sites implies homomolecular vicinity, i.e., that $NH_3$

molecules easily are nearest neighbours of other ammonia ones. If one of those molecules receives a proton from a nearby water molecule, Coulombic repulsion then hinders the formation of other ammonium cation in its first neighbour shell, which is the case illustrated in Fig. 5d. At the end of our AIMD trajectory, we observe that nearly 90% of the neutral $NH_3$ molecules are surrounded by at least one $NH_4^+$ ion. Moreover, we find that Grotthuss-like water–hydroxide proton hopping are relatively common, in agreement with results in the literature[41–43], whereas ammonium–water and ammonium–ammonia proton hopping almost never occur. This last observation is consistent with the much more extreme thermodynamic conditions necessary to observe superionicity in pure ammonia ice[10], and suggests that ammonium ions are, at our milder conditions, "topological proton wells", which sequestrate Grotthuss-diffusing protons, thus hindering a more efficient charge distribution within the crystal, and, at last, to attain a full ionisation as in P4/nmm.

To sum up, the observed $Im\overline{3}m$ phase is topologically frustrated by homomolecular vicinity and, as a result, cannot be fully composed of ionic species, thus remaining partly molecular. The experimental observation that P4/nmm is only produced locally and as a minor phase likely results from the fact that the formation of P4/nmm requires a local configuration, where ammonia molecules are exclusively surrounded by water, or vice versa, a situation which has low probability. It is also likely that proton transfer between water and ammonia molecules precede and trigger the structural phase transition, and while the system tends towards the ordered ionic state, it is kinetically trapped into a lattice-topological frustrated state, dictated by the specific local arrangements, which permit, or not, proton transfer events from water to ammonia.

## Discussion

The present work shows that the ammonia monohydrate compound spontaneously converts into an unusual crystalline state at high pressure, where the standard molecular forms of water and ammonia coexists with hydroxyl and ammonium ions. The relatively low-pressure onset for static ionisation (7.4 GPa) in AMH contrasts with the extreme pressures required in the pure ices (150 GPa in $NH_3$[11], over 1.4 TPa in $H_2O$[44]) and can be understood by the lower energy cost for the proton transfer between water and ammonia.

Consistent with previous theoretical works[33, 34], the lowest enthalpy crystal for AMH is found to be the ordered and fully ionic P4/nmm phase in the pressure range 10–40 GPa. However, this phase is only obtained at best as a minor phase in all our experiments. On the other hand, our X-ray and neutron diffraction data are consistent with the cubic, highly disordered structure of ref. [29]. We solve this apparent riddle by showing that the cubic phase is not purely molecular, as described in ref. [29], but is a mixed ionic-molecular one (called DIMA), and that the system is trapped in this state and cannot reach its global minimum energy one. Due to the highly disordered nature of this phase, the present experiments alone do not allow to discriminate between the purely molecular and mixed ionic-molecular state, which made the input of ab initio calculations essential to understand this system.

Besides the fundamental aspects, AMH might be present in the interiors of icy bodies of the solar system, and these ionisation properties could be relevant in their modelling, in particular to understand the unusual magnetic fields of the icy planets Neptune and Uranus. Superionicity with large proton conductivity has been predicted in AMH in different structures at significantly higher P/T conditions[34], close to those of the respective pure ices[8, 10, 17]; we speculate, and leave that to future research, that proton mobility might set in at much milder conditions, and be significantly higher over the whole phase diagram.

## Methods

**Experiments**. Equimolar mixtures of water and ammonia were prepared in a stainless steel cylinder as in refs. [28, 29]. Ammonia gas (99.99% purity) obtained from Air Liquide and deionised water were used. Ammonia was condensed at liquid–nitrogen temperature into a preweighed stainless steel bottle. The bottle was sealed, warmed and the amount of ammonia condensed measured by weighing the bottle again. The correct mass of water to create a 1:1 molar mixture was put into a second steel bottle, which was again sealed and the two bottles were connected together. The bottle containing the water was cooled in liquid nitrogen while the other, containing the ammonia, was held at room temperature. The valves sealing the bottles were opened and the ammonia condensed into the cold bottle containing the water ice. The valves were then closed and the mixture was allowed to warm to RT and homogenise for over 24 h. The final composition was checked by weighing the mixed solution and comparing to the sum of the masses of the two components. The difference was in the range 0.08–1.74%, and whenever it was larger than 1%, the mixture was discarded and prepared again.

For DAC loading, the DAC body was cooled to ~100 K using liquid $N_2$ and the equimolar mixture was cooled to ~210 K using a isopropanol-ℓ-$N_2$ mixture. A drop of the liquid was then poured on the gasket hole, which froze in contact with the DAC body and was compressed to a pressure between 10 and 25 GPa before letting the DAC warm to room temperature. The compression and warming rates, as well as the pressure at which the sample is warmed up, were not precisely controlled and varied between loadings. According to ref. [29], the stable solid phase recovered using this procedure is AMH-VI, whereas the direct RT compression of the liquid results in a mixed AHH + ice sample[28]. Rhenium gaskets were used in all experiments. Pressure was determined from the wavelength shift of ruby fluorescence[45] and temperature below 300 K were measured using a diode sensor attached to the cold head of the cryostat.

Fourier transform infrared absorption experiments were performed either on the SMIS beamline of the SOLEIL synchrotron facility (Saint-Aubin, France), which provides a 25 μm beam in the mid-IR, or with an in-house interferometer, with a 120 μm beam in the mid-IR. IR spectra were collected in the range 600–8000 $cm^{-1}$. Synthetic type IIa diamond anvils (Almax industries) with flat culets of 300 or 400 μm diameter were used. In order not to saturate the absorption, the sample was loaded on top of KBr pellet to produce a thin sample film of ~1 μm. The reference spectra were obtained with the DAC loaded with pure KBr.

Raman spectra were collected with an in-house spectrometer using the 514.5 nm line of an argon laser from spectra physics focussed to a spot of ~2 μm on the sample. The backscattered light was filtered by razor edge filters from Semrock and dispersed by a HR450 spectrograph from Horiba coupled to a CCD camera from Andor.

Angular dispersive X-ray diffraction experiments were done at the European Radiation Synchrotron Facility (ESRF), beamline ID27 ($\lambda = 0.3738$ Å), or using an in-house diffractometer based on a rotating Mo anode from Rigaku. Diffraction images were recorded on 2D detectors (MarResearch marCCD at ESRF, Rigaku RAXIS-IV at IMPMC). At the ESRF, the scattered X-rays were filtered by a multichannel collimator to reduce the large contribution originating from the Compton scattering of the diamond anvils[46]. Integration of the X-ray images was performed with the fit2D[47] or Dioptas[48] software programs. High-pressure neutron-diffraction experiments were carried out using the V7 Paris-Edinburgh press on the PEARL station of the ISIS spallation neutron source at the Rutherford Appleton Laboratory.

**Ab initio random structure searching**. We used the AIRSS[40] method in order to predict the most promising crystal phases at given pressures. These ab initio random structural searches generate and relax random structures to a minimum in the enthalpy. At the most basic level, the method aims to construct unit cells with random lattice vectors and place atoms in the chosen stoichiometry at random positions. The lattice vectors and atomic positions are fully relaxed to the local minimum using DFT, and the enthalpies compared to find the global minimum. The AIRSS searches were used with the CASTEP plane wave code[49] with ultrasoft pseudopotentials[50] and the Perdew–Burke–Ernzerhof (PBE)[51] functional. For the searches, we used a plane wave cutoff energy of 340 eV and a Monkhorst–Pack (MP)[52] Brillouin zone sampling grid of $2\pi \times 0.07$ Å$^{-1}$. In order to relax the generated structures at a higher level of accuracy, we employed a cutoff energy of 500 eV with a Brillouin zone sampling of $2\pi \times 0.03$ Å$^{-1}$.

We first carried out searches using 2 and 4 formula units (f.u.) of $NH_3 \cdot H_2O$ per unit cell at pressures of 10, 20, 30 and 40 GPa. Additional searches with f.u. of ions ($NH_4 \cdot OH$) were performed in order to bias the searches. We found several different relaxed structures, either molecular, ionic and mixed ionic-molecular but, the lowest enthalpy structures in all 2 and 4 f.u. searches up to 40 GPa is the ionic P4/nmm as space group as found by Griffiths et al.[33] at lower pressures and confirmed in a recent theoretical study[34] using a different method of structural search.

**Theoretical Raman and IR spectra**. We used the PHONON code of the Quantum Espresso[53] package to calculate the vibrational modes and corresponding infrared and Raman intensities using the density functional perturbation Theory[54]. Ultrasoft pseudopotentials[50] were adopted to describe the electron–ion interaction within the PBE gradient correction scheme[51]. A 1360 eV cutoff energy was used and Brillouin zone integrations were performed at the $\Gamma$ point.

**Classical and ab initio molecular dynamics simulations**. We constructed ten $4 \times 4 \times 4$ supercells ($a = 3.325$ Å, which corresponds to the experimental pressure of about 10 GPa) containing 64 $NH_3.H_2O$ f.u. (448 atoms), and one $6 \times 6 \times 6$ supercell with 216 f.u. (1512 atoms). To obtain substitutional-disordered configurations, O and N atoms were distributed randomly on each site of the bcc lattice. Then, we placed hydrogen atoms in randomised molecular orientations.

Classical molecular dynamics simulations were performed using the GROMACS code[55]. Our simulations were performed in the NVT ($T = 300$ K) ensemble with a timestep of 0.5 fs and a simulation time equal to 5 ns using a rigid body potential. The potential energy function consists of Coulomb and Lennard–Jones terms using the following expression (Eq. 1):

$$V(r_{ij}) = \sum_i \sum_{j>i} \left( \frac{q_i q_j e^2}{r_{ij}} + \frac{A_{ij}}{r_{ij}^{12}} - \frac{B_{ij}}{r_{ij}^6} \right) \quad (1)$$

Molecules are represented by interaction sites ($i$ and $j$ in Eq. 1) usually located on the nuclei. The $A_{ii}$ and $B_{ii}$ parameters are given by $A_{ii} = 4\epsilon_{ii}\sigma_{ii}^{12}$ and $B_{ii} = 4\epsilon_{ii}\sigma_{ii}^6$, where $\sigma_{ii}$ and $\epsilon_{ii}$ are the Lennard–Jones radius and well depth. For $A_{ij}$ and $B_{ij}$, $i \neq j$, we used the relations $A_{ij} = \sqrt{A_{ii}A_{jj}}$ and $B_{ij} = \sqrt{B_{ii}B_{jj}}$. The N–H bonds are fixed to be equal to 1.010 Å with HNH angle of 106.40. For water, we set the O–H bond at 0.960 Å and HOH angle at 104.52. The parameters ($q$, $\sigma$ and $\epsilon$) for $NH_3$ and $H_2O$ are those given in refs. [56, 57], respectively.

Density functional theory calculations were performed with the boxes at fixed volume in order to optimise the atomic coordinates. We used the PW code of the Quantum Espresso package[53]. Ultrasoft pseudopotentials were adopted to describe the electron–ion interaction within the PBE gradient correction functional. A kinetic energy cutoff of 680 eV was used on the wavefunctions, and 5440 eV for the charge density. Due to the use of supercells, only the $\Gamma$ point was taken into account in the Brillouin zone. At convergence, we observed in all 10 supercells the formation of ionic species with concentrations between 8 (12%) and 11 (17%) f.u. of $NH_4.OH$. We also optimised the atomic positions for the supercell containing 1512 atoms and a total of 29 (13%) f.u. of $NH_4.OH$ were formed. This indicates that the concentration of ionic pairs is not much dependent on the size of the supercell. The average energy of the different configurations of 448 atoms is equal to $-788.288 \pm 0.008$ eV fu$^{-1}$. The difference in energy between the $P4/nmm$ structure and the simulated boxes, at the same volume per f.u., is around 0.3 eV fu$^{-1}$. This high-energy difference points out the topological frustrated property of the DIMA phase.

Ab initio (Born–Oppenheimer) molecular dynamics was employed to carry out finite temperature simulations ($T = 300$ K) with a timestep of 0.38 fs. The system was equilibrated at RT in the NVT ensemble during 2 ps using a Berendsen thermostat[58]. After this equilibration, the system contained 17 units of $NH_4·OH$ and 47 units of $NH_3·H_2O$. Following equilibration, the thermostat was removed, and the simulation was performed in the microcanonical (NVE) ensemble for 20 ps.

**Data availability**. The data that support the findings of this study are available from the corresponding authors on reasonable request.

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

## Acknowledgements

We thank M. Mezouar for assistance in the XRD experiments at ESRF. We acknowledge financial support from the Agence Nationale de la Recherche under grants ANR-15-CE30-0008-01 (SUPER-ICES) and ANR-13-BS04-0015(MOFLEX). This work was also supported by French state funds managed by the ANR within the Investissements d'Avenir programme under reference ANR-11-IDEX-0004-02, and more specifically within the framework of the Cluster of Excellence MATISSE led by Sorbonne Universités. We acknowledge the synchrotron SOLEIL for provision of beam time allocated to proposals 20120199 and 20130489 and the ESRF for provision of beam time allocated to proposals HC-1646 and HC 2483. We acknowledge the GENCI (IDRIS and CINES) French national supercomputing facilities for CPU time (Project 091387 2015 and 2016).

## Author contributions

S.N., F.D. and A.M.S. designed the project. C.L. and A.M. equally contributed to the paper. C.L., J.A.Q., C.W., H.Z., K.B.,G.L.M., B.B., P.D., G.G., J.S.L., F.D. and S.N. performed the experiments. A.M. and A.M.S. performed the computer simulations. C.L., J.A.Q., F.D. and S.N. analysed the experimental data. A.M., F.P. and A.M.S. analysed the numerical data. F.D., S.N., A.M. and A.M.S. wrote the paper. All authors discussed the results and commented on the manuscript.

## Additional information

**Competing interests:** The authors declare no competing financial interests.

