## [Peer Review file · Nature Communications]

Reviewers' comments:

Reviewer #1 (Remarks to the Author):

Liu. et al reported an interesting piece of high pressure work about water-ammonia mixture. Experimentally, they measured Raman/IR of the sample at 12 GPa and XRD/ND at 12 GPa (the same pressure). With Raman measurement, they claimed that they observed OH⁻ stretching and NH₄⁺ bending modes, which has good agreement with previously suggested a purely ionic P4/nmm H₂O:NH₃ compound. With XRD, they observed Im-3m molecular AMH with a minor peaks suggesting the existence of P4/nmm ionic compound. I cannot recommend to publish it to Nature comm. while they are touching a hot field of research area because of the following reasons.

Vibrational spectrum is an indirect measure of the structures. A similarity of peak positions cannot be a rigorous reason to claim the structure. From figure 1, the best I can agree with is the observation of OH⁻ stretching mode, which is a unique behaviour. However, experimental data are not clearly taken and there is much room the two peaks near 1500 cm⁻¹ can be interpreted as other vibration modes than NH₄⁺. In connection with this, the data in figure 3 is not clear either. In my opinion, it seems the peaks near 1500 cm⁻¹ still exists down to 4.9 GPa while OH⁻ stretching modes disappear below 7.4 GPa.

XRD shows a clear evidence of formation of molecular AMH. Authors claim the a minor phase of P4/nmm by magnifying the main peak near (110), which is marginal to be believed.

As a backup, they conducted ab initio calculations. Structure searching only reproduced the result from the previous literature [33]. Authors tried to explain Im-3m construction from systematic atomic displacement from P4/nmm, which is not a scientific approach. Actually this implies that Im-3m is not energetically favourable within DFT scheme at zero temperature. They might need to consider to include temperature effect (Gibbs free energy). The discussion part after this is purely speculation without rigours evidence.

Overall, the the authors' research subject is interesting enough, however, their logic described the paper is not sufficient

Reviewer #2 (Remarks to the Author):

This paper presents results on the partial ionization of water/ammonia mixtures under high pressure conditions. The results should be of interest to the high pressure community and the planetary science community. The partial ionization is an interesting effect that has not been previously highlighted.

The quality of the experimental investigations is very high. The researchers use a variety of state of the art diagnostics to gain insight into the disordered state of the material.

I am not sure if "topological frustration" is the best way of describing the author's results. If we consider a simple acid-base reaction between ammonia and water, one would expect an ionic/neutral equilibrium in liquids based on the pK_a. The authors have shown that under high pressure condition, the pK_a shifts to favor ionization, but that neutral species are still present in the equilibrium.

Therefore I do not believe that results should be described as an exotic state of matter (called disordered ionic-molecular crystal), but should instead be described a partially ionized amorphous solid, or amorphous/crystalline mixture. The authors should refer to concepts of acid/base chemistry, which aptly describes the observations without invoking exotic states of matter.

I recommend publication after the authors revise the language used to describe their results to

remove reference to novel states of matter, and instead describe their results in terms of equilibrium between ionized and non-ionized states in solids.

Reviewer #3 (Remarks to the Author):

A review report for the manuscript entitled

“Topologically frustrated ionisation in a water-ammonia ice mixture”

authored by

C. Liu, A. Mafety, J.A. Queyroux, C. Wilson, K. Beneut, G. Le Marchand, B. Baptiste, P. Dumas, G. Garbarino, F. Finocchi, J.S. Loveday, F. Pietrucci, A.M. Saitta, F. Datchi, and S. Ninet

prepared by Taras Palasyuk

The manuscript reports on very interesting results obtained for ammonia – water system under high pressure studied by complementary experimental (Raman, IR, XRD, ND) and a number of theoretical methods.

The main claim of the manuscript – the experimental observation of partial self-ionization taking place between ammonia and water molecules, which leads to formation NH_4^+ and OH^- at relatively mild pressure conditions.

It would be definitely interested for a broad range of reader audience.

In general the manuscript is well structured and written, however, somewhat chaotic bringing about confusion for the reader.

The manuscript, in my opinion, needs certain improvement/clarification before publishing. Further evidence/analysis, which will definitely provide more grounds to make conclusions convincing includes:

1. In general, a thorough description of starting ammonia – water mixtures is missing in the manuscript. It is difficult to draw clear conclusions on homogeneity of mixtures being measured. Please provide information on homogeneity of samples after being loaded at low temperature and after recovery to room temperature. A presentation of each initial sample characterisation with various techniques along with comparison to known data from the literature would be highly advisable.
2. A table (see below) including a brief description of conditions used for sample preparation along with a concise presentation of results obtained by particular experimental technique would be much useful in both analysis and ability to reproduce the work. If it is possible fill out the table for all 30 sample studied. Feel free to add entries which are missing from the table and, in your opinion, could provide valuable details.

#	Loading conditions	Loading procedure	Measurement procedure	Experimental data

					I o b s e r v a t i o n s				
	Temperature	Pressure		Pressure increase		Pressure unloading run	Raman	IR	XRD

3. Please provide XRD raw images and calibration details (for fit2D) for samples where, in opinion of the authors, the reflection originating from P4/nmm structure of ionic compound (NH₄OH) are the most intense.
4. Presentation of experimental data in the full pressure range (up to 40 GPa) would be advisable.
5. Analysis of pressure effect on hydrogen bonding and the role of particular hydrogen bonds (both homo- and heteronuclear) are missing in the manuscript. In that respect, plots presenting an evolution of both Raman and IR spectra as well as the position of particular Raman/IR modes under pressure would be much useful.
6. The argument presented on the Page 11 regarding strong Coulomb repulsion due to close contacts of ions of the same polarity seems to be a hasty generalization till there is no clear picture of dispersive interactions, e.g. hydrogen bonds, provided.
7. Please provide input crystallographic data used for calculation of vibrational properties of P4/nmm phase and supercell calculations.
8. It would be useful to draw any conclusion on the pressure evolution of ionic phase abundance: any plausible estimation on how the molecular/ionic phase ratio is changing with pressure.

Text editing remarks:

1. The references do not comply the format of Nature Communications
2. Page 16, the Paragraph "Ab Initio Random Structure Searching" , in the line "... CASTEP plane wave code [?]" reference is missing.

Yours faithfully,

Dr Taras Palasyuk

Response to Referee 1

- *Referee 1 notes : « Vibrational spectrum is an indirect measure of the structures. A similarity of peak positions cannot be a rigorous reason to claim the structure. From figure 1 (Nota: we believe Referee 1 actually refers to figure 2), the best I can agree with is the observation of OH-stretching mode, which is a unique behaviour. However, experimental data are not clearly taken and there is much room the two peaks near 1500 cm⁻¹ can be interpreted as other vibration modes than NH₄⁺. In connection with this, the data in figure 3 is not clear either. In my opinion, it seems the peaks near 1500 cm⁻¹ still exists down to 4.9 GPa while OH- stretching modes disappear below 7.4 GPa.»*

In the manuscript (pp 4-6 and Fig. 2) we compare the experimental Raman and IR spectra of AMH at 12 GPa to those predicted by DFT for the P4/nmm structure. On one hand, we observe that all the bands predicted for P4/nmm are present in the experimental data, but on the other hand, the data also clearly show vibrational bands that do not belong to P4/nmm and have frequencies typical of H₂O and NH₃ molecules. The agreement in frequency between the observed and predicted bands of P4/nmm, including for the Raman lattice modes which are very sensitive to the crystal structure, is in our opinion a strong indication of the presence of this phase in the sample. We have added to the revised version of the manuscript the Supplementary figure 2 presenting the pressure dependence of the vibrational bands, which shows that the agreement between the predicted and observed

frequencies extends to the full pressure range probed by our experiments. But the occurrence of molecular H₂O and NH₃ vibrations show that the sample is not purely composed of P4/nmm since the latter is a fully ionic structure. This is also supported by our XRD data which can be interpreted as a mixture of the Im-3m and P4/nmm phases (see also below).

Concerning the IR bands around 1500 cm⁻¹, we agree that the spectral changes across the transition at ~7 GPa are not as easy to see as those occurring at 3700 cm⁻¹ in Fig. 3. This is because the former region contains both vibrations from NH₄⁺ twisting, NH₃ twisting and H₂O bending, while only OH⁻ stretching occurs at 3700 cm⁻¹. There are however clear changes in the 1450-1600 cm⁻¹ spectral window between 7.4 GPa and 6.8 GPa, going from a doublet peaked at 1460 and 1550 cm⁻¹, which coincide with the predicted frequencies for NH₄⁺ twisting in P4/nmm, to a multiplet ranging from 1480 to 1660 cm⁻¹, which can be explained by pure NH₃ and H₂O bands. These changes are clearer when one looks at the evolution of the peak frequencies vs pressure, which we have included as the Supplementary Figure 2 in the revised version of the manuscript. The multiplet observed at and below 6.8 GPa resembles the one measured for the low-P, low T molecular phases of AMH I, as shown in Fig 3, which is actually true for all the internal vibration bands (the external ones are different since the lattice of the HP phase is different from AMH-I).

- *Referee 1 notes : "XRD shows a clear evidence of formation of molecular AMH. Authors claim the a minor phase of P4/nmm by magnifying the main peak near (110), which is marginal to be believed."*

As stated in the manuscript (pp 7-8), the XRD pattern shown in Fig. 4 is dominated by the peaks from the Im-3m structure. Peaks from the P4/nmm structure are much weaker and only appear as side bands of the (110) peak of Im-3m in the integrated pattern. In the revised version we have added the Supplementary figure 3 which presents the 2D XRD image and two different integrations: using the complete image on one hand, as done in Fig. 4, or using a integration box centered on the partial rings which can be indexed by P4/nmm, in order to better emphasize the latter peaks.

Our work also shows that the Im-3m structure can be equally described as a molecular or ionic or partially ionic structure without significant incidence on the quality of the fit to the XRD and neutron diffraction data. As our calculations suggest, the Im-3m structure is not purely molecular as previously thought, but contains both neutral and ionic species.

- *Referee 1 notes : "As a backup, they conducted ab initio calculations. Structure searching only reproduced the result from the previous literature [33]. Authors tried to explain Im-3m construction from systematic atomic displacement from P4/nmm, which is not a scientific approach. Actually this implies that Im-3m is not energetically favourable within DFT scheme at zero temperature. They might need to consider to include temperature effect (Gibbs free energy). The discussion part after this is purely speculation without rigours evidence."*

Ab initio calculations are an integral part of this study and represent a substantial amount of work. They were particularly important in order to distinguish molecular from ionic models of the Im-3m structure, which as recalled above, was difficult to resolve from the experimental data alone. It is true that our structure searches confirm the result of Ref. [33] that P4/nmm is the lowest enthalpy structure at 10 GPa, but it also extends the latter work by showing that P4/nmm remains the lowest enthalpy structure in the searches up to 40 GPa. As stated in the manuscript, the orientational and substitutional disorder of the Im-3m structure makes it intrinsically impossible to be found in ab initio random searches which are typically limited to 8 formula units. We also emphasize that this disorder "precludes a direct description of the Im-3m structure in theoretical calculations, and thus to discuss its stability with respect to P4/nmm".

The fully ionic Im-3m model which we discuss in the manuscript, obtained by orientationally and substitutionally disordering the P4/nmm structure, was actually proposed by the authors of Ref. [33]. We state in the manuscript that, although this model fits equally well the diffraction data, we think that it is not physically reasonable since it implies that two ammonium ions have a 25 % probability to be first neighbors and the associated energetic cost due to Coulomb repulsion would be very large.

Having established that the Im-3m structure is the dominant one in our samples and that it coexists with the fully ionic P4/nmm structure, we try and answer the question whether the Im-3m structure is purely molecular as initially described by Ref. 29. This is the purpose of the extensive calculations described in the last part of the manuscript. Since Im-3m cannot be directly simulated as said above, we used approximants containing 128 randomly-placed molecules (448 atoms) in a 4x4x4 supercell of the bcc unit cell. We considered 10 of those cells to detect possible bias in the random choice of molecular positions. We also considered a 6x6x6 supercell (432 molecules, 1512 atoms) to look for size effects. After optimization at 0 K, all cells showed a spontaneous but limited ionization (from 12 to 17 % depending on the simulation cell) of the molecules. We show that the ionization of a molecule is dictated by its local environment, which explains the partial ionization of the simulation box. Temperature effects at 300 K were indeed included in the simulations using AIMD, which showed that the ratio of ionized molecules increased from 12 % at 0 K to ~36 % at 300 K because the thermally-activated reorientation of the molecules enabled additional proton transfers from a water molecule to an ammonia one.

We cannot agree that these theoretical results are pure speculation as they are based on well optimized and well converged state-of-the-art calculations. Most importantly, the results draw a physically reasonable model of the system which is compatible with all experimental observations.

Response to Referee 2

- *Referee 2 notes: « This paper presents results on the partial ionization of water/ammonia mixtures under high pressure conditions. The results should be of interest to the high pressure community and the planetary science community. The partial ionization is an interesting effect that has not been previously highlighted. The quality of the experimental investigations is very high. The researchers use a variety of state of the art diagnostics to gain insight into the disordered state of the material.»*

We thank the Referee for pointing out the interest and novelty of our results, and the high quality of our investigations.

- *Referee 2: « This I am not sure if "topological frustration" is the best way of describing the author's results. If we consider a simple acid-base reaction between ammonia and water, one would expect an ionic/neutral equilibrium in liquids based on the pKa. The authors have shown that under high pressure condition, the pKa shifts to favor ionization, but that neutral species are still present in the equilibrium. Therefore I do not believe that results should be described as an exotic state of matter (called disordered ionic-molecular crystal), but should instead be described a partially ionized amorphous solid, or amorphous/crystalline mixture. The authors should refer to concepts of acid/base chemistry, which aptly describes the observations without invoking exotic states of matter.»*

This remark is quite pertinent, and we have carefully considered the Referee's interesting "chemistry-based" point of view. However, we believe that the concepts of acid/base chemistry and

ionic/molecular equilibrium, which hold for the liquid phase, cannot be extrapolated to the present solid ice mixture. Indeed, on one hand, all known solid forms of AMH which are stable below 7 GPa are molecular solids solely composed of water and ammonia molecules. It is likely that ionic point defects exist in these solids as in the pure ices but only at very low concentration ($[H_3O^+] < 10^{-13}$ in water ice at 253 K according to Petrenko & Whitworth, "Physics of ice", p. 154). The ambient pressure liquid mixture, according to the CRC Handbook of chemistry, has a pKa of 9.25, meaning that there are about 1% ammonium ions. This is much more than the expected concentration of ionic defects ($< 10^{-13}$) in the molecular crystal phase (AMH-I) into which the liquid crystallizes at low temperature.

On the other hand, the number of $NH_4^+.OH^-$ ionic pairs in the DIMA phase at high pressure reaches, according to our simulations, about 36% at 300 K, which is much more than in the liquid phase at the same temperature. There is no data for the pKa of liquid AMH at high pressure but the ab initio simulations of Ref. 34 indicate that the liquid remains mainly molecular below 25 GPa at 1000 K. We also note that the high pressure DIMA phase, despite being highly disordered, remains a crystal with long-range order as evidenced by the presence of Bragg peaks in the x-ray and neutron diffraction patterns.

Furthermore, ab initio calculations carried out in this work and by other groups before indicate that the thermodynamically stable state of AMH above 7 GPa is the fully ionic P4/nmm structure. There is no equivalent fully ionic solution in ammonia/water mixtures, again suggesting that acid/base chemistry arguments which holds for liquid mixtures cannot be extrapolated to the solid ices. To illustrate this in a different way, consider a crystalline supercell, as done in the manuscript, where ammonia molecules sit at the corner of an elementary BCC lattice, and water molecules at the cell center. When this system is relaxed using first-principles calculations at 10 GPa, *all* water molecules spontaneously (i.e. barrierless) cede a proton to a nearest-neighbor ammonia one, in stark contrast with the ionic/molecular chemical equilibria in the liquid solution.

When substitutional disorder is introduced in the crystal, as in the Im-3m DIMA phase, our calculations reveal that ionization systematically occurs when proton-acceptor ammonia molecules have no previously-formed ammonium ions as nearest neighbours, and systematically does not occur in the opposite case, which is what we call "topological frustration". We do agree with the referee that the notion of "exotic state" could be subjective in this case, and have replaced this expression by "unconventional form".

Response to referee 3:

- *Referee 3 notes "In general, a thorough description of starting ammonia – water mixtures is missing in the manuscript. It is difficult to draw clear conclusions on homogeneity of mixtures being measured. Please provide information on homogeneity of samples after being loaded at low temperature and after recovery to room temperature. A presentation of each initial sample characterisation with various techniques along with comparison to known data from the literature would be highly advisable. »*

The preparation of the starting mixture is described in the Methods section. It is identical to the one used in Ref. [28,29], but following the Referee's request we have added the detailed operation in the revised version. Water and ammonia are known to be fully miscible in the liquid phase at any concentration. After mixing the two components, the mixture is given sufficient time to homogenize at room temperature, usually over 24 hrs. We also take the precaution to shake the sample bottle before loading. The final composition was checked by measuring the mass of the sample and

comparing it with the sum of the masses of the two components: the difference was in the range 0.08-1.74%. Whenever this difference was over 1%, the mixture was discarded and prepared again.

To check that our method provides the correct 1:1 composition, we also performed XRD of one sample at ambient pressure and 140 K using a laboratory powder diffractometer (with Cu anode) equipped with a cryostat. The pattern was typical of the AMH-I crystal phase with the addition of a few percent ice Ih coming from the condensation of residual water inside the cryostat sample chamber. Note that the water ice cannot come from the sample itself since for concentrations deviating from 1:1 the sample would be AMH-I+AHH-I or AMH-I+ADH-I depending whether the NH₃ content is larger or smaller, respectively. No Bragg peaks from water ice could be observed in the DAC samples studied by XRD, showing that ice did not condense inside the sample hole during loading. The neutron pattern of the deuterated sample taken at near ambient pressure in the neutron experiment was also typical of AMH-I and no water ice peaks was observed in this experiment too.

During loading, the mixture is cooled to ~210 K, which is close to the freezing temperature (194 K) and ensures a low vapour pressure (below 0.2 bar) and thus prevents from changes in concentration by loss of the more volatile ammonia component. The DAC body is cooled to ~100 K, which ensures a rapid freezing of the liquid when it gets into contact with the cell. The cell is then rapidly closed and the sample compressed.

As explained in the text, in all DAC experiments, sufficient load was applied at low temperature in order to achieve a pressure higher than 10 GPa, where AMH is in phase VI at RT. This procedure was employed in order to avoid the dehydration of AMH which occurs upon compression of the liquid at RT, as reported in Ref. 28.

The homogeneity of the sample at 12 GPa and RT is discussed in the manuscript based on the Raman and x-ray diffraction results. Both methods suggest that the sample is a mixture of the Im-3m (AMH-VI) and P4/nmm phases. As said in p. 8, the XRD patterns are always dominated by the reflections of Im-3m, indicating that this phase is dominant in all samples. Bragg peaks from P4/nmm are observed at several positions of the sample with small intensity (see the new Supplementary figure 3). In p. 6 we also indicate that “the intensities of the Raman peaks assigned to the P4/nmm ionic structure depend on the position of the laser spot (of size 2-3 μm) on the sample and are anti-correlated to those assigned to the molecular species, suggesting that the distribution of ionic species is not homogeneous over the size of the sample. Moreover, these peaks varied in intensity between different loadings, and could not be observed at all in some samples.”

- *Referee 3 notes : « A table (see below) including a brief description of conditions used for sample preparation along with a concise presentation of results obtained by particular experimental technique would be much useful in both analysis and ability to reproduce the work. If it is possible fill out the table for all 30 sample studied. Feel free to add entries which are missing from the table and, in your opinion, could provide valuable details. »*

As said above, the Methods section has been expanded to include more details on the sample preparation. All samples were prepared in the same way and either studied by Raman, IR spectroscopy or by XRD (or a combination of these techniques). Only one (deuterated) sample was studied by neutron diffraction. The setups used for these experiments are also described in the Methods section in details and are quite standard. We will be happy to reply to any other questions readers may have on our experimental methods.

- *Referee 3 notes : "Please provide XRD raw images and calibration details (for fit2D) for samples where, in opinion of the authors, the reflection originating from P4/nmm structure of ionic compound (NH₄OH) are the most intense. »*

The revised version includes the Supplementary figure 3 showing a XRD image where the two more intense reflections of the P4/nmm structure are clearly visible. Since these are only partial rings, the integration of the full image gives a pattern where these reflections only appear as shoulders of the (110) peak of Im-3m, as in Fig. 4 of the main text . We therefore included a different integration, using a box centered on the partial rings, which make the P4/nmm peaks more visible. We will be happy to provide the raw data on request after the paper is published.

- *Referee 3 notes : « Presentation of experimental data in the full pressure range (up to 40 GPa) would be advisable. »*

The revised version includes the Supplementary figures 1 and 2 which show the experimental data over the full pressure range. We did not include them in the main text as our manuscript is centered on the discussion of the nature of the observed high pressure phases, not their evolution with pressure.

- *Referee 3 notes : « Analysis of pressure effect on hydrogen bonding and the role of particular hydrogen bonds (both homo- and heteronuclear) are missing in the manuscript. In that respect, plots presenting an evolution of both Raman and IR spectra as well as the position of particular Raman/IR modes under pressure would be much useful »*

The plots requested by the Referee have been included in the revised version as Supplementary figures 2 and 3. A thorough description of hydrogen bonds and their evolution under pressure is beyond the goal of this study which rather focusses on the pressure-induced self-ionisation of the sample and the topological frustration resulting from the substitutional disorder. Clearly though, these bonds are important as the ionisation events observed in our simulations always occur from a donor water molecule to an acceptor ammonia molecule which are H-bonded, according to a Grotthus mechanism, as illustrated in Fig. 5b and 5c. We have added this information in the text, p.12 . On the other hand, we note that the substitutional and orientational disorder of the DIMA phase makes the identification of the role of H-bonds on the vibrational modes a very complex task.

- *Referee 3 notes : « The argument presented on the Page 11 regarding strong Coulomb repulsion due to close contacts of ions of the same polarity seems to be a hasty generalization till there is no clear picture of dispersive interactions, e.g. hydrogen bonds, provided »*

We agree that the initial version was not accurate enough on this point, as the Coulomb repulsion mostly concerns the neighbouring ammonium ions and not the hydroxide ions which can bond together via H-bonds, as actually observed in our simulations. We have changed the text to clarify this point.

- *Referee 3 notes : « Please provide input crystallographic data used for calculation of vibrational properties of P4/nmm phase and supercell calculations ».*

We have added the calculated P4/nmm structural parameters at 12 GPa in the table 1 of the SM. The vibrational density of states for the simulated pseudo Im-3m phase with 448 atoms was obtained from the trajectories of the AIMD simulations at 300 K. This cannot easily be presented in a table. We will be happy to provide cif files giving the atomic positions at the end of the simulation for this phase on request.

- *Referee 3 notes : « It would be useful to draw any conclusion on the pressure evolution of ionic phase abundance: any plausible estimation on how the molecular/ionic phase ratio is changing with pressure. »*

The molecular ionic/ratio appears not to be much affected by pressure as the relative intensity of the IR bands from ionic and molecular species do not change with pressure (see figure 3 of the main text and figure 1 of the SM). We have added this remark in the main text (p. 6). A likely explanation for this constant ratio is that, as our simulations show, ionization of a molecule depends on its local environment, which does not change with increasing pressure as long as no structural phase transition occurs.

Text editing remarks:

1. *The references do not comply the format of Nature Communications*

We have changed the references to comply with the format of Nature Communications.

2. *Page 16, the Paragraph “Ab Initio Random Structure Searching” , in the line “... CASTEP plane wave code [?] ” reference is missing.*

The missing reference has been included in the revised version.

Reviewers' comments:

Reviewer #1 (Remarks to the Author):

First, I should acknowledge that the authors work hard to support their report in this revised manuscript. Personally I tempted to be convinced because it contains an interesting suggestion of NH₃ and H₂O interaction at moderate pressure and temperature range within very tractable range of contemporary high pressure techniques, however, several points are not clear for me. One of the key issue is that how much can we be sure of the existence of P4/nmm phase of AMH based on their XRD figure. (200) and (111) peaks are shown as minor peaks in the Figure 4 (a) inset, however, it was not well reproduced in their repeated XRD and it is not observed in ND experiment.

Again here, I would like to stress it out that Raman is an indirect measure of the structure but XRD shows a rather clear clue. Considering this, XRD result, if it is a clear result, should appear at first in their manuscript. Raman shift at 3745 cm⁻¹, strictly speaking, is a property of a local geometry of OH⁻ not P4/nmm AMH. Another AMH with OH⁻ unit and a different structure can possess the same Raman feature. Anyhow, we can see OH⁻ modes in their work, however, it is not the evidence of ionic ammonia unit. NH₄⁺ vibrational information in the Figure 2 is very limited; Lattice modes are not clear, NH₄⁺ twisting is not clear, NH₄⁺ stretching seems too broad. The only match is NH₄⁺ bending near 1500 cm⁻¹. Actually, this frequency gave me another inspiration that the sample might reacted to the environment such as gasket. What is the gasket material? Is it Rhenium? Then, ReH_x formation can explain the experimental data. It was reported to be formed above 5 GPa (T. Atou, 1995) and mono hydrides have a vibration near 1500 cm⁻¹ and thus H₂O + Re interaction might occur, for example. I couldn't find information of the gasket material in their text.

DFT result found P4/nmm should be stabilized around this pressure. It is another interesting point why there is a contradiction between theory and experiment. Authors used this contradiction as a starting point to claim "topological frustration". However, there are many possible scenarios such as "van der Waals correction", "entropy induced structural transformation". Using a classical force field calculation, they annealed supercell and relaxed it using DFT. Most likely, it can end up with local minima and it is unlikely to be believed the relaxed structure is a representing one of the experimental system.

Another Reviewer also raised the point questioning the term "topological frustration" in AMH and I agree with her/him. Whether P4/nmm phase is real, I cannot make a connection this phenomena to a topological problem. Rather, it is like to be originated from disorderedness of the system. In their AIMD, they also claimed that Coulomb interaction between NH₄⁺ units prevent a fully ionic phase. What's the reason to omit OH⁻, which screens the Coulomb interaction?

At current stage, I am not convinced based on their manuscript and thus I cannot recommend it to be published.

Reviewer #2 (Remarks to the Author):

The authors have provided a very thorough response to my previous letter, and have met all of my concerns. I recommend that the manuscript be published as is.

Reviewer #3 (Remarks to the Author):

The points raised in my previous review has been explicitly addressed. In my opinion, there is no major objection for publishing the revised manuscript.

Response to Referee 1

Referee 1 writes: *"One of the key issue is that how much can we be sure of the existence of P4/nmm phase of AMH based on the their XRD figure. (200) and (111) peaks are shown as minor peaks in the Figure 4 (a) inset, however, it was not well reproduced in their repeated XRD and it is not observed in ND experiment. Again here, I would like to stress it out that Raman is an indirect measure of the structure but XRD shows a rather clear clue. Considering this, XRD result, if it is a clear result, should appear at first in their manuscript. Raman shift at 3745 cm⁻¹, strictly speaking, is a property of a local geometry of OH⁻ not P4/nmm AMH. Another AMH with OH⁻ unit and a different structure can possess the same Raman feature."*

Even though we are convinced that our data give strong evidence of the presence of P4/nmm in our samples, we believe this is not the central result of our study. Indeed this phase was previously predicted to be the stable one at high pressure so its experimental observation was rather expected. More interesting in our opinion is the experimental fact that the major phase in all our samples is the Im-3m one, not P4/nmm, and our efforts have been rather focussed on understanding the nature of the Im-3m phase and why it is preferred over P4/nmm. The key result is that, unlike previously thought, the Im3m phase hosts both molecular and ionic species and that the ionization process, which would be complete in P4/nmm, is topologically frustrated in Im-3m as a result of the substitutional disorder of this phase.

Regarding the identification of a crystalline phase by Raman and IR spectroscopy: it is well known that the vibrational spectrum of a crystalline material, whether probed by Raman or infrared spectroscopy, is strongly linked to its crystalline structure, a fact which has long been used to determine the presence of a crystal phase in a sample. As said in the manuscript and repeated in our previous response, all the Raman and IR bands expected for the P4/nmm structure are clearly observed in the experimental spectra shown in Fig. 2, at frequencies close to those predicted by DFT. This includes the molecular modes, such as the OH⁻ stretching and NH₄⁺ bending and twisting, AND the lattice modes. The only modes which are difficult to discern are the NH₄⁺ stretching ones because they overlap with the broad stretching modes of H₂O and NH₃. The fact that we observe the correct number of Raman lattice modes at frequencies very close to those predicted gives a strong indication that the observed phase is P4/nmm. We agree that the stretching OH⁻ vibration in P4/nmm could incidentally coincide with that of another ionic structure, and cannot be used alone as a proof of P4/nmm. Still, the fact that we observe a single peak constrains the symmetry of the OH⁻ sites and shows that there is a single OH⁻ site in the unit cell as in P4/nmm.

As stated in the manuscript, the P4/nmm phase is at best observed in minor quantity in all our samples and is not homogeneously distributed. The reasons why P4/nmm is more easily observed by spectroscopy rather than XRD or ND comes in our opinion from the facts that (1) the Raman and IR bands of P4/nmm are sharp and intense, while those of Im-3m are broad and weaker (2) We use a confocal Raman system with a tightly focussed laser beam, allowing to collect the scattered light from a small volume of the sample and thus select a region with higher P4/nmm content (3) The XRD and ND probes are less local as we collect the diffracted signal from the entire sample thickness and over a larger section (4) the higher symmetry of Im-3m concentrates the diffracted intensity in fewer Bragg peaks compared to P4/nmm (5) the P4/nmm phase is more textured than Im-3m and only small fractions of diffraction rings are observed.

Referee 1 writes: *"Anyhow, we can see OH⁻ modes in their work, however, it is not the evidence of ionic ammonia unit. NH₄⁺ vibrational information in the Figure 2 is very limited; Lattice modes are not clear, NH₄⁺ twisting is not clear, NH₄⁺ stretching seems too broad. The only match is NH₄⁺ bending near 1500 cm⁻¹. Actually, this frequency gave me another inspiration that the sample might reacted to the environment such as gasket. What is the gasket material? Is it Rhenium? Then, ReH_x formation can explain the experimental data. It was reported to be formed above 5 GPa (T. Adou, 1995) and mono hydrides have a vibration near 1500 cm⁻¹ and thus H₂O + Re interaction might occur, for example. I couldn't find information of the gasket material in their text."*

Since the Referee appears not to object to the presence of OH⁻, it is difficult for us to understand his/her concerns about the presence of NH₄⁺. A positively charged ion is undoubtedly required to keep the electric neutrality of the sample, and it is unlikely that an isolated H⁺ ions would have a long lifetime in the sample considering its strong affinity to bond to ammonia molecules. Moreover, we clearly observe the signature of the Raman NH₄⁺ twisting and IR NH₄⁺ bending vibrations.

The Referee suggests a possible reaction between the sample and Re gasket (we do use Re gasket, this information has been added to the Methods section), which could explain the 1500 cm⁻¹ band. He/She cites the work of T. Adou & J.V. Badding (J. Sol State Chem 118, 2999, 1995) who reported the formation of ReH_x, x=0.38(4), above 5 GPa at 300 K. We note that the latter was obtained by the reaction between Re powder in contact with molecular H₂ and it is indeed well known that molecular H₂ easily diffuses into metals at high pressure, as discussed for example in Datchi et al, PRB 61,6535 (2000). We respond that, first, there is no evidence for the formation of H₂ in the bulk of our sample, either by Raman, IR or XRD. Second, if ReH_x was formed, it would be localized at the edge of the gasket, as observed in previous works (eg, Degtyareva et al, Sol. State Comm. 149, 1583, 2009). Our gasket holes were typically of 100-120 μm diameter at 10 GPa and the Raman, IR and XRD beams which we used in our experiments were 4 to 50 times smaller than this, so we could not be observing the signal from the gasket. Third, the ReH_x phase is most likely metallic and our samples remained transparent at any investigated P-T conditions. Fourth, we could not find any experimental report of the Raman spectra of ReH_x, and thus wonder why the Referee states there should be a band at 1500 cm⁻¹. Fifth, once formed, ReH_x is stable down to ambient pressure (see Adou and Badding's paper), while we clearly see the Raman and IR peaks from OH⁻ and NH₄⁺, including the modes around 1500 cm⁻¹, disappear and replaced by H₂O and NH₃ modes on decompression below 7.4 GPa (see Figure 3 of the manuscript).

The above arguments hold for any other possible reaction between the sample and gasket, be it Re oxide or hydroxyde. In any case, the reaction would be limited to the edge of the gasket and not visible at the center of the sample where we probe it. We further note that no such reaction has to our knowledge been reported in any of the numerous experimental studies of pure water up to megabar pressures at either room or high temperature. The same is true for pure ammonia which we have extensively studied in the past up to extreme P-T conditions using Re gaskets (see Refs 10, 11 and 39 of the manuscript).

Referee 1 writes: *"DFT result found P4/nmm should be stabilized around this pressure. It is another interesting point why there is a contradiction between theory and experiment. Authors used this contradiction as a starting point to claim "topological frustration". However, there are many possible scenarios such as "van der Waals correction", "entropy induced structural transformation". Using a classical force field calculation, they annealed supercell and relaxed it using DFT. Most likely, it can end up with local minima and it is unlikely to be believed the relaxed structure is a representing one of the experimental system."*

As stated in the manuscript and our previous response to Referee 1, the codes used for structural prediction can only deal with ordered structures and a small number of formula units per unit cell. This makes it impossible to find the $Im\bar{3}m$ phase. Since we cannot directly compare the enthalpy of $P4/nmm$ and $Im\bar{3}m$ using DFT, there is no definite answer on whether there is a contradiction between theory and experiment.

Our simulations show that if we start from a supercell where water are exclusively surrounded by ammonia molecules and vice-versa, the system spontaneously evolves towards the fully ionic $P4/nmm$ phase. By contrast, in a substitutionally disordered crystal such as the $Im\bar{3}m$ phase, the ionization is only partial which we explain by the fact that there is a finite probability that 2 ammonia molecules be first neighbours and that Coulomb repulsion inhibits the formation of 2 ammonium ions at such close distance. The topological distribution of water/ammonia molecules on the crystalline sites inherently generates local molecular configurations, which are thus the determining factor for the ionisation process, which is why we refer to the $Im\bar{3}m$ phase as topologically frustrated.

Our experimental data constrains the symmetry of the major phase to be body-centered cubic, which imposes substitutional disorder on each lattice site. We use this experimental knowledge as an input for our simulations. As stated in our previous response, the latter include temperature, and thus entropic effects since we run molecular dynamics simulations at 300 K. Eventually, we obtain a model which satisfies all experimental observations and is physically sound, so it is reasonable to think it is representative of the real system.

Referee 1 writes: *Another Reviewer also raised the point questioning the term “topological frustration” in AMH and I agree with her/him. Whether $P4/nmm$ phase is real, I cannot make a connection this phenomena to a topological problem. Rather, it is like to be originated from disorderedness of the system. In their AIMD, they also claimed that Coulomb interaction between NH_4^+ units prevent a fully ionic phase. What’s the reason to omit OH^- , which screens the Coulomb interaction?*

Referee 2 proposed to view the coexistence of ionic and molecular species as a chemical equilibrium similar to liquid solutions rather than a topological frustration. We responded to this in our previous revision and, in view of Referee’s 2 reply, convinced him that the concept of topological frustration is better adapted to the present case. We thus refer Referee 1 to this response.

Indeed, the topological frustration originates from the substitutional disorder of the system, as stated in the manuscript and repeated above. We did not omit OH^- ions, these simply cannot screen the Coulomb interaction between 2 ammonium ions which are first neighbours. By contrast, as said in our previous response to Referee 3, the hydroxide ions can bond together via H-bonds if they are properly oriented, so they can be first neighbours, which is actually observed in our simulations.

The paper reports the formation of (i) a partially-ionized $Im\bar{3}m$ molecular phase and (ii) an ionic $P4/nmm$ phase in AMH at relatively low pressures.

Hereafter I detail the major critics:

- Raman and IR peaks varied in intensity between different loadings and could not be observed at all in some samples. The authors should explain the lack of reproducibility.
- The single Raman and IR peak at $\sim 3750\text{ cm}^{-1}$ suggests a single OH- site. The X-ray diffraction figure only shows two minor extra peaks. From these evidences it can not be unequivocally inferred the existence of the tetragonal $P4/nmm$ ionic structure.
- The authors claim that they measured another sample, where the diffraction peaks of the $P4/nmm$ were more intense, and the (200) and (111) reflections of this phase appear more clearly together with other additional peaks of this structure. Why is not shown?
- How do the authors explain the appearance of the low-temperature phase I in the decompression measurements at RT? Figure 3.
- There is no signal or evidence of the $P4/nmm$ phase from time-of-flight powder neutron diffraction. Why? The scattering strength of deuterium should provide precise H(D) atoms locations.
- In the simulations, the authors randomly distribute water and ammonia molecules using bcc supercells. Then, they relaxed the structure by DFT. Which symmetry, lattice parameters and atomic coordinates have the optimized partially ionic pseudo-bcc structures? I guess the final partially-ionized molecular crystals can not longer be described within the cubic $Im\bar{3}m$ symmetry. Are the relaxed pseudo-bcc structures compatible with the XRD and neutron diffraction patterns? This is a critical point. How compare energetically the most stable supercell structures to the $P4/nmm$ structure?

In conclusion, the manuscript is currently not clear for the reader and the existence of the partially-ionized $Im\bar{3}m$ molecular phase and the ionic $P4/nmm$ phase in AMH is not unequivocally demonstrated. Results from experiments and simulations are not convincing. In my opinion, this manuscript should not be published in Nature Communications unless these major concerns are addressed.

Response to Referee 4:

- 1. Referee 4 :** *“Raman and IR peaks varied in intensity between different loadings and could not be observed at all in some samples. The authors should explain the lack of reproducibility.”*

Our experiments show that the variability in the Raman peak intensities is related to the variable amount of the P4/nmm phase in the sample. The latter is always in minor proportion and is non-homogeneously distributed in the sample, the dominant phase being always I-m3m. As said in the manuscript, we believe this is due to the fact that, in P4/nmm, ammonia molecules are exclusively surrounded by water, and vice versa, a situation which has low probability to occur. Indeed the samples are produced by compression at low temperatures (~100 K), going in particular through solid phases I and II, where no such local environment exists. Furthermore, the compression and warming rate during loading, as well as the final pressure, were not precisely controlled and thus varied between loadings. There is also a small variability (<1%) in the composition of the initial mixture. This variability is likely to have an incidence on the number of ionic species and on the probability to nucleate one phase or the other, as our simulations suggest that the formation of a OH⁻/NH₄⁺ ionic pair is strongly dependent on the local topology. Finally, kinetic effects such as slow molecular diffusion and large potential barriers are likely responsible for the fact that once formed, the system is trapped in the disordered bcc phase and cannot reach the energetically more favorable P4/nmm structure.

- 2. Referee 4:** *“The single Raman and IR peak at ~3750 cm⁻¹ suggests a single OH⁻ site. The X-ray diffraction figure only shows two minor extra peaks. From these evidences it cannot be unequivocally inferred the existence of the tetragonal P4/nmm ionic structure. The authors claim that they measured another sample, where the diffraction peaks of the P4/nmm were more intense, and the (200) and (111) reflections of this phase appear more clearly together with other additional peaks of this structure. Why is not shown? »*

The diffraction pattern for the second sample was obtained with a lab source and was not of good quality, which is why we chose not to show it. For the present revision, we have collected new diffraction patterns of this sample at the ESRF synchrotron. The latter unambiguously shows the presence of the P4/nmm phase as several peaks are observed. The strong texture prevents from doing a Rietveld refinement, but a good Le Bail fit was achieved using a mixture of Im-3m and P4/nmm phases, as seen in the new Fig. 4. For consistency, we also changed the Raman spectrum in Fig. 2 to the one collected from this sample (which is very similar to the initial one) and relocated the Raman spectra and XRD pattern shown in previous versions to the supplementary figure 3.

- 3. Referee 4:** *“How do the authors explain the appearance of the low-temperature phase I in the decompression measurements at RT? Figure 3”*

As specified in the caption of Figure 3, the spectrum of the low temperature phase I was reproduced from Ref. 35 and does not come from the RT decompression of our sample. Our purpose was to show that the phase obtained on decompression below 7.4 GPa has IR modes

in the same frequency windows as the molecular phase I. This and the absence of the peak at 3750 cm⁻¹ strongly suggest that the sample reverts to a purely molecular below 7.4 GPa.

- 4. Referee 4:** *“There is no signal or evidence of the P4/nmm phase from time-of-flight powder neutron diffraction. Why? The scattering strength of deuterium should provide precise H(D) atoms locations.”*

There are slight differences in the loading procedures used for the neutron experiments compared to the DAC experiments: the compression is made at higher temperature, 150 K, and the warming rate of the 50kg neutron cell is slower (it takes several hours compared to less than an hour for DAC). Based on the arguments given in 1), these differences likely have an incidence on the nucleation of the P4/nmm phase. The neutron pattern shown in previous versions presented an asymmetric (110) peak which could be due to the presence of P4/nmm but the evidence was not significant enough. For this revision, we looked more carefully at data taken with other samples and found that on occasions the non-bcc peaks appear more clearly. The present revision thus include a neutron pattern with stronger evidence for P4/nmm (figure 4(b)).

- 5. Referee 4:** *“In the simulations, the authors randomly distribute water and ammonia molecules using bcc supercells. Then, they relaxed the structure by DFT. Which symmetry, lattice parameters and atomic coordinates have the optimized partially ionic pseudo-bcc structures? I guess the final partially-ionized molecular crystals can not longer be described within the cubic Im-3m symmetry. Are the relaxed pseudo-bcc structures compatible with the XRD and neutron diffraction patterns? This is a critical point. How compare energetically the most stable supercell structures to the P4/nmm structure?”*

Clearly, ionisation locally breaks the Im-3m symmetry and distorts the cubic unit cell. For this reason, no symmetry is imposed to the simulation cells during optimization. But when averaged over many unit cells, as diffraction does, the Im-3m symmetry is conserved. To show this, the diffraction pattern of the optimized 6x6x6 cell (216 fu) is compared to the neutron pattern at 10 GPa in the new figure 5a. It can be seen that the computed pattern agrees very well with experiment. In particular, the calculations reproduce the rapid decrease of the peak intensities when d decreases, which is clearly related to the large disorder of the structure. The same level of agreement is obtained for the x-ray pattern, shown in supplementary new figure 4.

As indicated in the Methods section, the difference in enthalpy given by DFT between P4/nmm and the average of the optimized bcc supercells with 448 atoms is 0.3 eV/f.u at 0 K. This large energy difference clearly shows that the partially ionized bcc phase is not the ground state at low temperature, and that a large kinetic barrier prevents the system to go from Im-3m to P4/nmm.

REVIEWERS' COMMENTS:

Reviewer #4 (Remarks to the Author):

Overview

The manuscript shows for the first time experimental evidences (Raman, IR, XRD and neutron diffraction results) of the existence of the fully ionic AMH $P4/nmm$ structure, which was previously found stable in theoretical works. The authors also suggest the existence of a highly disordered ionic-molecular phase (DIMA) from their ab initio calculations, but their experiments do not allow to discriminate that phase from a pure molecular cubic $Im-3m$ structure. The existence of a mixed ionic-molecular phase in AMH remains therefore unproved.

Previous concerns

I am satisfied with the revisions and answers to my comments.